# Protective Effects of Black Rice Anthocyanins on D-Galactose-Induced Renal Injury in Mice: The Role of Nrf2 and NF-κB Signaling and Gut Microbiota Modulation

**DOI:** 10.3390/nu17030502

**Published:** 2025-01-29

**Authors:** Dan Sun, Wei Wang, Qian Fan, Xinyi Wang, Xinyan Xu, Weiye Jiang, Liang Zhao, Hong Li, Zhifang Fu, Lei Zhao, Hongmei Jiao

**Affiliations:** 1Department of Geriatrics, Peking University First Hospital, Beijing 100034, China; 13581525356@163.com (D.S.); lih7510@163.com (H.L.); fuzhifang2004@126.com (Z.F.); 2Beijing Key Laboratory of Functional Food from Plant Resources, College of Food Science & Nutritional Engineering, China Agricultural University, Beijing 100083, China; 13856012304@163.com (W.W.); xaesi23@163.com (X.W.); 3Key Laboratory of Geriatric Nutrition and Health, Beijing Technology and Business University, Ministry of Education, Beijing 100048, China; fanqian@st.btbu.edu.cn (Q.F.); xuxy2023@163.com (X.X.); jiangwyjy@163.com (W.J.); liangzhao@btbu.edu.cn (L.Z.)

**Keywords:** aging, kidney, D-galactose, black rice anthocyanins, Nrf2, NF-κB

## Abstract

Background/Objectives: This study aimed to evaluate the renal protective effects of black rice anthocyanins (BRAs) against renal injury in mice induced by D-galactose (D-gal). Methods: The renal aging mouse model was established by thirteen consecutive weeks of subcutaneous injections of D-gal. The serum levels of urea nitrogen (BUN), creatinine (CRE), uric acid (UA), antioxidant enzymes (e.g., GSH-Px and SOD), and total antioxidant capacity (T-AOC), as well as the contents of inflammatory factors (IL-1β, IL-6, and TNF-α) in kidney tissues were evaluated. Additionally, the relative expression of the NQO1, HO-1, IKKβ, NF-kBp65, and TLR4 proteins was examined. Results: BRA treatment significantly reduced serum levels of BUN, and CRE increased the concentrations of antioxidant enzymes and total antioxidant capacity in renal tissues, and reduced the levels of inflammatory factors. Furthermore, BRAs restored the relative expression of the NQO1, HO-1, IKKβ, NF-kBp65, and TLR4 proteins to normal levels and promoted the recovery of the renal tissue architecture. Conclusions: It was demonstrated that BRAs could potentially prevent and protect against kidney injury by modulating the Nrf2 and NF-κB signaling pathways, attenuating oxidative stress and inflammatory responses, and modulating the gut microflora. These findings provide a scientific basis for the application of BRAs as a natural bioactive substance in the field of nephroprotection, especially against the renal degeneration that accompanies the aging process.

## 1. Introduction

Aging is an unavoidable physiological phenomenon in which the body’s functions gradually deteriorate with age, thereby greatly increasing the risk of disease and death [1]. Currently, kidney diseases caused by aging are a major public health burden [2]. Therefore, it is crucial that we prevent and treat kidney damage caused by aging. Nowadays, chemical drugs are mainly used for the prevention and management of kidney-related diseases, but they usually have certain side effects [3]—for example, angiotensin-converting enzyme inhibitors (ACEIs), angiotensin II receptor blockers (ARBs), and statins [3,4], etc. Research indicates that the increased prevalence of kidney injury among the elderly population can primarily be attributed to the synergistic effects of aging and pharmacological intervention [5]. Íris Guerreiro et al. showed that plant polyphenols and their metabolites can participate in the signaling cascade of the oxidative stress response, anti-inflammatory mechanisms, and apoptosis, and have great potential in improving kidney-related diseases [6]. Studies have also shown that natural bioactive substances such as resveratrol, curcumin, puerarin, and anthocyanins possess therapeutic potential in alleviating acute kidney injury and diabetic kidney disease injury [7,8,9,10]. For instance, Kšonžeková, Milutinović, and colleagues demonstrated that the anthocyanin-rich extract of chokeberry waste significantly alleviated cisplatin-induced acute renal injury in rats by reducing oxidative stress and inflammatory responses [9]. Longobardi et al. found that anthocyanin-rich red orange and lemon extract exerted a protective effect against ochratoxin A-induced renal injury in rats by modulating the Nrf2 signaling pathway [10]. These results highlight the potential of anthocyanins as therapeutic and preventive agents for kidney protection.

Black rice anthocyanins (BRAs), a polyphenol derived from black rice, are naturally active substances with antioxidant, anti-inflammatory, and anti-aging properties [11,12,13]. Previously, numerous studies have explored the anti-aging effects of BRA, such as extending the lifespan of *Drosophila melanogaster*, and improving the conditions of aging mice [14,15]. Additionally, BRAs have been reported to improve kidney injury. Qi et al. showed that BRAs can prevent diabetic nephropathy in rats, mainly by preventing renal insufficiency and renal fibrosis [16]. Feng et al. demonstrated that BRAs can alleviate hyperuricemia in mice, primarily by reducing uric acid levels and inhibiting key enzymes in the uric acid production pathway, thereby inhibiting uric acid production. This intervention also reduced kidney injury and oxidative stress in mice [17]. Nonetheless, the existing literature regarding the impact of BRAs on aging-related kidney injury is relatively sparse, indicating a need for more comprehensive investigations. Therefore, it is crucial that we study the role of BRAs in aging-induced kidney injury, not only to reduce the incidence of kidney-related diseases but also to mitigate other health problems caused by the long-term use of therapeutic drugs.

Numerous studies have established that the administration of D-galactose (D-gal) is a prevalent approach for establishing aging models that mimic natural aging, and this model is extensively utilized in research focused on combating age-related degeneration and organ damage [18]. Consequently, the current study utilized a murine aging model, which was established through the subcutaneous injection of D-gal into the dorsal cervical region of mice. We measured body weight, organ coefficients, relevant physicochemical indices, histological lesions, and gut microorganisms to investigate the renal protective effect of BRAs against D-gal-induced renal injury in mice. Furthermore, we investigated the influence of BRAs on the expression of pertinent proteins within the Nrf2 and NF-κB signaling pathways using a Western blot analysis, to elucidate the potential mechanisms by which BRAs mitigate renal injury associated with aging.

## 2. Materials and Methods

### 2.1. Materials

The D-gal utilized in this investigation was procured from Beijing Boao Tuoda Technology Co., Ltd. (Beijing, China). Black rice anthocyanin (BRA, HPLC = 94.59%) was obtained from Nanjing Jingzhu Biotechnology Co., Ltd. (Nanjing, China). Commercial kits for the determination of urea nitrogen (BUN), creatinine (CRE), uric acid (UA) in serum, superoxide dismutase (SOD), total antioxidant capacity (T-AOC), malondialdehyde (MDA), glutathione peroxidase (GSH-Px), and reduced glutathione (GSH) in renal tissue homogenates were supplied by Nanjing Jiancheng Bioengineering Institute (Nanjing, China). In addition, commercial kits for measuring inflammatory factors in serum, including interleukin-1β (IL-1β), interleukin-6 (IL-6), and tumor necrosis factor-α (TNF-α), were provided by Lianke Biotechnology Co., Ltd. (Hangzhou, China). Antibodies against Nrf2, NQO1, HO-1, MyD88, and Keap1 were acquired from Wuhan Proteintech Group, Inc. (Wuhan, China). TLR4, IKKβ, NF-kBp65, NF-kBp50, and IKBα antibodies were provided by Hangzhou Huaan Biotechnology Co., Ltd. (Hangzhou, China). All measurements were conducted in triplicate, and all procedures were performed following the manufacturers’ instructions for the respective kits. All additional chemicals and reagents utilized in the study were of analytical-grade quality.

### 2.2. Animal Experiments

In this study, a total of forty male C57BL/6J mice, aged between 6 to 8 weeks, were procured from Charles River Laboratories (Beijing, China), under License No.: SCXK (Beijing) 2021-0006. All experimental protocols received approval from the Ethics Committee for Animal Experiments at Beijing Technology and Business University (Approval No. Lunshen 2024 No. 163). All experimental animals were housed under standardized conditions with a 12 h light/dark cycle, at a controlled temperature of 22 ± 2 °C and relative humidity of 50 ± 10%. Prior to the commencement of the experiment, the mice underwent a 7-day acclimatization period to facilitate their adaptation to the experimental environment. After the acclimatization period, the mice were randomly assigned to four groups: the control group (C), the D-gal group (M), the low-dose BRA administration group (L-BRA), and the high-dose BRA administration group (H-BRA), with each group comprising 10 mice. Subsequently, the aging model was induced using D-gal [19]. During the modeling process, the three experimental groups received daily subcutaneous injections of D-gal at a dosage of 500 mg/kg at the nape of the neck, whereas the control group was administered an equivalent volume of saline (10 mL/kg) subcutaneously [20]. Concurrently, the mice in the low-dose and high-dose BRA groups were administered BRA via gavage at dosages of 50 mg/kg and 100 mg/kg, respectively [17]. All procedures were conducted at 9:00 AM and lasted for 13 weeks. Serum samples from mice were collected 24 h following the last administration, with all procedures adhering to the 3Rs principles to minimize distress. The serum samples were subsequently stored at −20 °C until further analysis. Additionally, kidney tissues and gut contents were collected from the mice and preserved at −80 °C for future studies.

### 2.3. Determination of Renal Function

After collecting blood samples from mice, they were refrigerated at 4 °C overnight. On a subsequent day, the samples underwent centrifugation at 3500 rpm for a duration of 20 min, after which the supernatant was carefully collected and preserved at −20 °C for subsequent analysis. The contents of creatinine (CRE), blood urea nitrogen (BUN), and uric acid (UA) in serum were quantified using biochemical assay kits. The manufacturer’s protocol for the assays was strictly followed to guarantee the accuracy and reproducibility of the results obtained.

### 2.4. Determination of Antioxidant Indexes

Kidney tissue samples (0.1 g) were supplemented with 0.9 mL of phosphate buffer solution (PBS). The tissues were homogenized with a mechanical homogenizer under ice bath conditions until completely pulverized. Then, the homogenates were centrifuged at 3500 rpm for 15 min at a temperature of 4 °C. The supernatant was then collected for the subsequent determination of biochemical indices. The contents of superoxide dismutase (SOD), total antioxidant capacity (T-AOC), malondialdehyde (MDA), glutathione peroxidase (GSH-Px), and glutathione (GSH) within the tissues were quantified utilizing the corresponding biochemical assay kits. The assays were strictly performed by the kit instructions to ensure the accuracy and reliability of the experimental results.

### 2.5. Measurement of Inflammatory Factors

The concentrations of interleukin-1β (IL-1β), interleukin-6 (IL-6), and tumor necrosis factor-α (TNF-α) in serum were quantified using specialized assay kits. All testing operations were strictly guided by the instructions provided with the kits to guarantee the precision and consistency of the results obtained.

### 2.6. Determination of Renal Histopathology

Following the euthanasia of the mice via cervical dislocation, the kidney tissues were quickly removed and immersed in a 4% paraformaldehyde solution for 24 h to preserve the structural integrity of the tissues. Thereafter, the tissue samples underwent dehydration through a series of progressively concentrated alcohol solutions to eliminate water content in preparation for subsequent embedding procedures. The dehydrated tissues were embedded in paraffin to facilitate sectioning. A microtome was used to slice the tissue into 5 µm-thick sections to facilitate microscopic observation. Finally, the sectioned samples were stained with conventional hematoxylin–eosin (HE) staining, which was used to assess tissue structure and cellular properties.

### 2.7. Determination of Gut Microorganisms

Mouse gut contents were collected and preserved at −80 °C for preservation. The samples were subsequently dispatched to Shanghai Majorbio Bio-pharm Technology Co., Ltd. (Beijing, China) for 16S rDNA gene sequencing under dry ice conditions. The sequencing experimental procedure mainly included steps of DNA extraction, PCR amplification, fluorescence quantification, library construction, and online sequencing. First, the genomic DNA was isolated by agarose gel electrophoresis. Then, PCR amplification of the designated sequencing region was performed using specific primers with barcodes, and the number of cycles was strictly controlled to maintain data accuracy. The amplified products were recovered and eluted from the gel, followed by fluorescence quantification to ascertain the sequencing quantity. Subsequently, the sequencing library was constructed, including the steps of adding splice sequences, recovering PCR products, and denaturing to produce single-stranded DNA fragments. Finally, the DNA fragments in the library were immobilized on the sequencing chip, and, after the steps of PCR synthesis, denaturation and annealing, bridge amplification, and single-stranded linearization, sequencing was performed using dNTP with fluorescent labeling, and the sequence information of the template DNA fragments was finally obtained through laser scanning and fluorescence signal collection. The relevant results obtained were analyzed on the Majorbio Cloud Platform (Website: https://cloud.majorbio.com/page/project/overview.html, accessed on 31 October 2024).

### 2.8. Western Blot Experiment

Twenty micrograms of kidney tissue were taken and mixed with PIPA lysis buffer supplemented with 1% protease inhibitors. The mixture was homogenized on ice with a homogenizer, and then centrifuged at 4 °C to collect the supernatant. An aliquot of this supernatant was used to quantify protein concentration by the BCA method. The samples were then mixed with 4× sampling buffer in a ratio of 3:1, heated in a water bath for five minutes, allowed to cool to room temperature, and either used immediately or stored at −80 °C. For SDS-PAGE electrophoresis, a 10% separation gel and a 5% stacking gel were prepared to facilitate the separation of proteins based on their molecular weights. Following electrophoresis, the proteins were transferred to a PVDF membrane and subsequently blocked with 5% skimmed milk powder for one hour at room temperature. The primary antibodies were diluted according to the manufacturer’s recommended ratios (β-actin at 1:50,000, Nrf2, Keap1, and TLR4 at 1:2000; and NQO1 at 1:20,000; HO-1, IKKβ, NF-kBp65, and IKBα at 1:1000) and incubated overnight at 4 °C. Secondary antibodies were diluted in 5% skimmed milk powder according to the corresponding ratio (β-actin, murine secondary antibody, and rabbit secondary antibody all at 1:50,000) and incubated for one hour at room temperature. Chemiluminescence was performed to develop the image. The grayscale values of the protein bands were analyzed using ImageJ software (Java 1.8.0), with the relative expression of the proteins being determined by the ratio of the grayscale values of the target protein bands of each group to the grayscale values of the bands of the internal reference β-actin. Statistical analyses were performed to evaluate the variations in protein expression levels.

### 2.9. Statistical Analysis

The quantitative data are expressed as mean ± standard deviation (SD). The significance of differences among factors was assessed using one-way ANOVA, accompanied by Duncan and LSD post hoc tests. All statistical analyses were conducted utilizing SPSS software, version 27.0. A *p*-value of less than 0.05 was deemed statistically significant.

## 3. Results

### 3.1. Effect of BRAs on Renal Function in Aging Mice Induced by D-gal

The impact of BRAs on renal function was evaluated through the measurement of serum concentrations of creatinine (CRE), blood urea nitrogen (BUN), and uric acid (UA). As illustrated in Figure 1, there was a statistically significant elevation in serum levels of BUN, CRE, and UA in the model group of mice when compared to the control group (*p* < 0.05). Following treatment with BRAs, a notable reduction in UA levels was observed in the high-dose BRA (H-BRA) group (*p* < 0.05) relative to the model group; however, no significant alteration was detected in the low-dose BRA (L-BRA) group (*p* > 0.05). Furthermore, no significant differences were identified in the levels of CRE and BUN between the BRA treatment groups and the model group (*p* > 0.05).

### 3.2. Effect of BRAs on Antioxidant Indices in Aging Mice Induced by D-gal

The impact of BRAs on kidney injury associated with aging, induced by D-gal, was evaluated through the analysis of superoxide dismutase (SOD), total antioxidant capacity (T-AOC), malondialdehyde (MDA), glutathione peroxidase (GSH-Px), and glutathione (GSH) levels in renal tissues. As illustrated in Figure 2, there were no statistically significant differences in the renal GSH and MDA concentrations between the D-gal model group and the control group, nor between the BRA-treated groups and the model group (*p* > 0.05). Conversely, the model group exhibited significantly lower levels of GSH-Px, SOD, and T-AOC compared to the control group (*p* < 0.05). Furthermore, no significant differences were detected in the levels of GSH-Px, SOD, and T-AOC between the L-BRA group and the model group (*p* > 0.05). In contrast, the H-BRA group demonstrated significantly elevated levels of GSH-Px, SOD, and T-AOC compared to the model group (*p* < 0.05), indicating a dose-dependent response.

### 3.3. Effects of BRAs on Serum Inflammatory Factors in Aging Mice Induced by D-gal

As illustrated in Figure 3, the serum levels of interleukin-1β (IL-1β), interleukin-6 (IL-6), and tumor necrosis factor-α (TNF-α) in the model group of mice were significantly elevated when compared to the control group (IL-1β and IL-6, *p* < 0.05; and TNF-α, *p* < 0.01). In contrast, both the L-BRA and H-BRA treatment groups demonstrated a significant decrease in the levels of IL-1β, IL-6, and TNF-α relative to the model group (L-BRA, *p* < 0.05; and H-BRA, *p* < 0.01), with the higher dosage exhibiting a more pronounced effect. Furthermore, when compared to the control group, the H-BRA group did not show a significant difference in IL-1β and IL-6 levels (*p* > 0.05), although the TNF-α level was significantly reduced (*p* < 0.05) and approached that of the control group.

### 3.4. Effects of BRAs on Renal Histology in Aging Mice Induced by D-gal

Figure 4 presents a histological evaluation of kidney tissues across various experimental groups. The control group had the most intact kidney tissue architecture, with minimal inflammatory reactions and abnormal cells, appearing essentially normal. In contrast, the model group showed a severe inflammatory response, marked by the extensive infiltration of inflammatory cells into the tissues and a loss of clarity in the structure of renal tubules and glomeruli, with substantial areas of damage. The L-BRA group revealed an irregular cell density in certain regions, alongside the presence of inflammatory cell infiltration and mild disorganization in the morphology of the local renal tissues. In contrast, the H-BRA group exhibited slight intercellular spacing and a limited degree of inflammatory cell infiltration; however, the overall architecture of the renal tissue remained largely intact.

### 3.5. Effect of BRAs on Nrf2 and NF-κB Signaling in the Kidneys of Aging Mice Induced by D-gal

In the context of the nuclear factor erythroid 2-related factor 2 (Nrf2) signaling pathway (refer to Figure 5A), the mice in the model group exhibited an increase in the relative expression of the kelch-like ECH-associated protein 1 (Keap1) protein in kidney tissues when compared to the control group; however, this increase did not achieve statistical significance (*p* > 0.05). Compared with the model group, the L-BRA group exhibited no significant change in the relative expression of Keap1 protein, indicating a comparable level. In contrast, the H-BRA treatment exhibited a significant reduction in the relative expression of Keap1 protein (*p* < 0.05). When comparing the model group to the control group, there was a noted downward trend in the relative expression of NAD(P)H:quinone oxidoreductase 1 (NQO1), Nrf2, and heme oxygenase-1 (HO-1) proteins in the kidney tissues of the model group, although these changes were not statistically significant (*p* > 0.05). Both L-BRA and H-BRA treatments significantly enhanced the relative expression of NQO1 and HO-1 proteins compared to the model group (*p* < 0.05). However, the effects of BRAs on the relative expression of Nrf2 protein were not significant (*p* > 0.05).

To the nuclear factor kappa-B (NF-κB) signaling pathway (Figure 5A), the model group exhibited a significant reduction in the relative expression of the inhibitor of NF-κB alpha (IKBα) protein in kidney tissues compared to the control group (*p* < 0.05). The L-BRA treatment group did not show any significant changes in the IKBα protein expression when compared to the model group. In contrast, the relative expression of the IKBα protein was significantly elevated in the H-BRA group (*p* < 0.05). Compared with the control group, the relative expression of IκB kinase beta (IKKβ), NF-κBp65, and toll-like receptor 4 (TLR4) proteins in the kidney tissues of mice in the model group was significantly elevated (*p* < 0.05). The L-BRA treatment did not significantly affect the relative expression of these three proteins. While the H-BRA treatment did not significantly alter the relative expression of the IKKβ and TLR4 proteins, it resulted in a significant reduction in the NF-kBp65 protein expression (*p* < 0.05).

### 3.6. Effects of BRAs on Gut Microflora of Aging Mice Induced by D-gal

The high dose was chosen to study the gut microflora, as this dose showed superiority across a spectrum of the aforementioned metrics. As shown in Figure 6A, at the species level, a substantial overlap of Operational Taxonomic Units (OTUs) was identified among the three experimental groups, with 190 OTUs being shared. Excluding these shared OTUs, there were 12 unique OTUs in comparison to the control group, while the H-BRA group presented 9 unique OTUs relative to the control group, and the model group had 11 unique OTUs when compared to the H-BRA group.

Table 1 indicates that there were no statistically significant differences in the diversity indices, specifically the Simpson, Shannon, Ace, and Chao indices, when comparing the model group to the control group (*p* > 0.05). In contrast to the model group, the H-BRA group displayed no significant alteration in Simpson’s index of diversity (*p* > 0.05). Nonetheless, a significant increase was noted in the Shannon, Ace, and Chao indices (*p* < 0.05). This suggests that the administration of H-BRA has a pronounced effect on increasing the diversity and abundance of gut microorganisms in aging mice subjected to D-gal-induced aging.

Figure 6B,C present the outcomes of the principal coordinate analysis (PCoA) at the OTU level, with a *p*-value of 0.006, indicating a significant difference in the gut microbial community structures among the groups. The model group demonstrated a clear divergence in its clustering compared to the control group. In contrast, the H-BRA group was distinctly clustered, separate from the model group, and its proximity to the control group suggests a similarity in the microbial community structure. Similarly, the cluster analysis, as depicted in Figure 6D, also showed that the H-BRA group shared a greater similarity in the gut microbial community structures of mice with the control group than the model group. This finding is in concordance with the outcomes of other analytical approaches, suggesting that H-BRA treatment may contribute to enhancing the diversity and richness of the gut microbiota in aging mice induced by D-gal.

As shown in Figure 6E, the gut microbial community composition at the phylum level was comparable across the three groups of mice, with Bacteroidota and Firmicutes emerging as the dominant phyla. However, significant variations in their relative abundances were observed. Specifically, the model group demonstrated a substantial increase in the relative abundance of Firmicutes, accompanied by a decrease in the relative abundance of Bacteroidota when compared to the control group. This resulted in a significant reduction in the ratio of Bacteroidota to Firmicutes (B/F) in the model group relative to the control group. Conversely, the H-BRA treatment led to a significant increase in the relative abundance of Bacteroidota and a decrease in Firmicutes, thereby enhancing the B/F ratio in comparison to the model group.

As illustrated in Figure 6F, at the genus level, the gut microbial community composition was essentially the same across the three groups of mice. However, notable variations in the relative abundances of specific genera were observed. Key genera, such as *norank_f_Muribaculaceae* and *Candidatus_Saccharimonas*, were consistently present in all groups. In comparison to the control group, the model group demonstrated a reduction in the relative abundance of *norank_f_Muribaculaceae* and an increase in that of *Candida-tus_Saccharimonas*. Furthermore, when compared to the model group, the H-BRA group showed an increase in the relative abundance of *norank_f_Muribaculaceae* and a decrease in the relative abundance of *Candidatus_Saccharimonas*.

By conducting a correlation analysis between the gut microbiota and key indicators such as renal function parameters, the antioxidant status, and inflammatory factors, the underlying role of the gut microbiota in the amelioration of age-related kidney damage by BRAs is further elucidated. As shown in Figure 6G, specific gut microbial genera showed significant negative correlations with key renal function indicators: *norank_f_Muribaculaceae* with BUN; *Dubosiella* with CRE; and *norank_f_Oscillospiraceae* with TNF-α.

These correlations combined with the changes in the relative abundance of gut microflora genera in the mice suggest that BRAs may attenuate age-related renal deterioration by enhancing the relative abundance of *norank_f_Muribaculaceae, Dubosiella*, and *norank_f_Oscillospiraceae*.

## 4. Discussion

The phenomenon of population aging is becoming increasingly pronounced, and, as individuals age, there is a concomitant decline in physiological functions, which contributes to a variety of health issues. Notably, the incidence of kidney-related diseases has risen significantly. Research indicates that the overall prevalence of chronic kidney disease (CKD) is projected to increase by 5.8% by the year 2027 in comparison to 2022 [21]. Consequently, it is imperative that we implement strategies for the prevention and protection against renal damage associated with aging. An increasing number of researchers are actively exploring effective natural compounds that may alleviate oxidative stress and inflammatory responses associated with aging. The current study focuses on examining the preventive and protective effects of BRAs against renal injury associated with aging, using a D-gal induced murine model to simulate age-related kidney damage.

Serum creatinine, urea nitrogen, and uric acid serve as critical biomarkers for evaluating renal health [22]. When renal function is compromised, these metabolites tend to accumulate in the bloodstream, leading to increased concentrations [22]. In this study, the administration of H-BRA resulted in a notable decrease in serum UA levels compared to the control group. Previous studies have established that age-related renal deterioration is often accompanied by abnormal fluctuations in these biomarkers [23,24]. Our findings indicated that BRAs effectively modulated serum UA levels, which aligns with research efforts to identify compounds that can delay aging and protect renal health. Furthermore, we analyzed the histopathological alterations in renal tissue utilizing HE staining. The findings indicated that D-gal induction caused significant structural damage to renal tissue [23]. Specifically, renal tissues from the model group exhibited marked alterations including cellular structural damages, degeneration, necrosis, and inflammatory cell infiltration. Previous research has indicated that renal injury associated with aging displayed similar histopathological characteristics. Notably, these pathological changes were significantly mitigated by the administration of BRAs, suggesting its beneficial reparative effects on renal injury induced by D-gal [23]. These results highlight the potential role of BRAs in preserving renal function associated with aging.

Research has also indicated that aging is a multifaceted and variable phenomenon closely linked to oxidative stress [23,24,25,26]. By assessing the antioxidant parameters in renal tissues, we found that the levels of SOD, GSH-Px and T-AOC were significantly elevated following H-BRA treatment [25]. In the context of aging-related renal injury, the generation of excessive free radicals by oxidative stress can cause significant damage to renal cells, thereby compromising their structural integrity and functionality. Elevated antioxidant levels can effectively neutralize these free radicals and alleviate renal damage [26]. The current study demonstrated that BRAs can counteract oxidative stress induced by D-gal. This finding is consistent with a previous report demonstrating that anthocyanins derived from *Dioscorea alata* L. and *Lycium ruthenicum Murr.* exhibited significant antioxidant activity in vivo [27,28]. These results suggest that BRAs hold significant potential as an antioxidant and in protecting the kidneys from oxidative-stress-related injury.

Chronic inflammation is another critical factor contributing to the aging process [29,30]. In this study, treatment with BRAs significantly reduced serum levels of inflammatory markers, including IL-1β, IL-6, and TNF-α. The persistent elevation of these inflammatory mediators in chronic inflammatory conditions can lead to renal tissue damage and expedite the aging process of the kidneys. The results of this study align with the renal changes observed in aging mice as reported by Lim et al. [31]. Additionally, findings from Nikbakht et al. and Zhang et al. indicated that anthocyanins exhibited anti-inflammatory properties in a dose-dependent manner [32,33]. Together, these findings highlight the potential of BRAs to mitigate age-related inflammation and provide renal protection.

The aging process is exacerbated by a reduction in Nrf2 levels and an increase in oxidative stress [34,35]. Nrf2 serves as a critical signaling pathway that modulates oxidative stress responses [30,36]. Under normal physiological conditions, Nrf2 is sequestered in an inactive form by binding to Keap1 [37]. In response to oxidative stress, the Nrf2–Keap1 complex dissociates, allowing Nrf2 to detach from the Keap1-mediated proteasome and translocate to the nucleus, where it binds to the antioxidant response element (ARE) to initiate the transcription of phase II antioxidant enzyme genes, specifically HO-1 and NQO1 [37,38]. The results of our study revealed that H-BRA significantly elevated the relative expression levels of NQO1 and HO-1 proteins while concurrently reducing the relative expression level of Keap1. This suggests that BRAs can activate the Nrf2 pathway, which is essential for maintaining the antioxidant capacity of renal cells in the context of aging-related renal injury [37,38]. Lee et al. have previously demonstrated the significant role of the Nrf2 pathway in mediating the antioxidant and anti-inflammatory effects of berry anthocyanins, which aligns with our findings [39].

As individuals age, a progressive decline in adaptive immunity is accompanied by the activation of innate immunity, contributing to the phenomenon of “inflammatory aging” or “inflammaging” [40]. The NF-κB signaling pathway serves as a critical regulator of innate immunity, modulating the expression of various inflammatory genes [41]. Under typical physiological conditions, NF-κB is retained in the cytoplasm by its inhibitors, known as IκBs [40,41]. However, stimulation with D-gal leads to the upregulation of TLR4, phosphorylation of IKKβ, activation of NF-κB, and subsequent degradation of IκB, thereby promoting inflammation [42,43]. Consequently, we examined the impact of BRAs on the degradation of TLR4, IKKβ, IκB-α, and NF-kB p65, as well as on NF-κB activation. Our findings indicated that BRAs significantly inhibited the degradation of IκB-α and suppressed NF-κB activation by markedly reducing the relative expression levels of the NF-kBp65 protein. In the context of aging-related renal injury, the excessive activation of the NF-kB pathway exacerbates inflammation and inflicts damage to renal tissues. Ferrari et al. have demonstrated that cyanidin-3-O-glucoside can mitigate chronic-inflammation-related diseases via the NF-kB pathway [44]. These findings align with our results, suggesting that BRAs may alleviate inflammation and confer protective effects against aging-related renal injury by inhibiting NF-kB pathway activation, thereby reducing inflammation.

Furthermore, the intestinal microbiota plays a crucial role in human health by regulating cellular growth and maintaining metabolic homeostasis [45]. Our results revealed a significant reduction in the intestinal microbial diversity in D-gal-induced aging mice, which was restored to baseline levels following treatment with BRA. It has been established that D-gal induced a decrease in the ratio of the phylum *Firmicutes* to the phylum *Bacteroidetes*, indicating a disruption in intestinal flora [46]. In the context of aging-related kidney injury, an imbalance in intestinal microbiota can indirectly impair renal function by affecting metabolic and immune functions. Compared to the model group, the BRA group exhibited an elevated ratio, approaching that of the normal group. This finding suggests that BRAs may exert nephroprotective effects by modulating the intestinal microbiota, although the specific mechanisms warrant further investigation. Numerous studies have highlighted the intricate relationship between the intestinal flora and renal health, and our research provided new evidence that BRAs ameliorated aging-related renal injury through the regulation of the intestinal microbiota. However, the potential molecular mechanisms and signaling pathways require further exploration.

Although this study demonstrated that BRAs prevented and protected against D-gal-induced kidney damage in mice, several limitations should be noted. First, only a specific mouse model was employed to imitate renal aging, which may not fully reflect the complexity of natural human aging. Additionally, the study included only a limited number of dosage groups and focused solely on the effects of short-term treatments. In terms of detection indices, the causal relationship between intracellular metabolic pathways and products, intestinal microorganisms, and renal injury were not thoroughly investigated. Moreover, potential interactions with other medications or chemicals that may affect renal function were not considered. Future research should address these limitations to provide a more comprehensive understanding of the nephroprotective effects of BRAs.

## 5. Conclusions

In summary, the current study found that BRA protects against kidney injury caused by aging. The potential mechanisms of action underlying these effects may involve enhancing the antioxidant capacity, attenuating inflammatory responses, regulating Nrf2 and NF-κB pathways, and modulating the gut microflora. These data suggest that BRA has the potential for the development of strategies aimed at preventing and mitigating kidney injuries induced by the aging process. These effects could be attributed to the increased antioxidant capacity, reduced inflammation, regulation of Nrf2 and NF-κB pathways, and modulation of the gut microbiota. These findings show that BRA has the potential to help create strategies for avoiding and treating kidney damage caused by the aging process.

## Figures and Tables

**Figure 1 nutrients-17-00502-f001:**
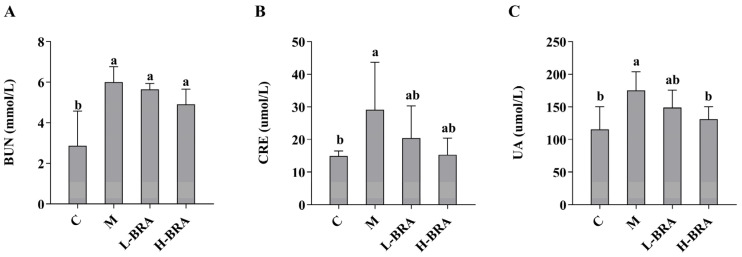
Effects of black rice anthocyanins (BRAs) on serum concentrations of urea nitrogen (BUN), creatinine (CRE), and uric acid (UA) in aging mice induced by D-galactose (D-gal). (**A**) BUN; (**B**) CRE; (**C**) UA. C: control group; M: D-gal model group; L-BRA: low-dose BRA group (50 mg/kg); and H-BRA: high-dose BRA group (100 mg/kg). The data are presented as the mean ± SD (*n* = 10). Different letters indicated significant differences (*p* < 0.05).

**Figure 2 nutrients-17-00502-f002:**
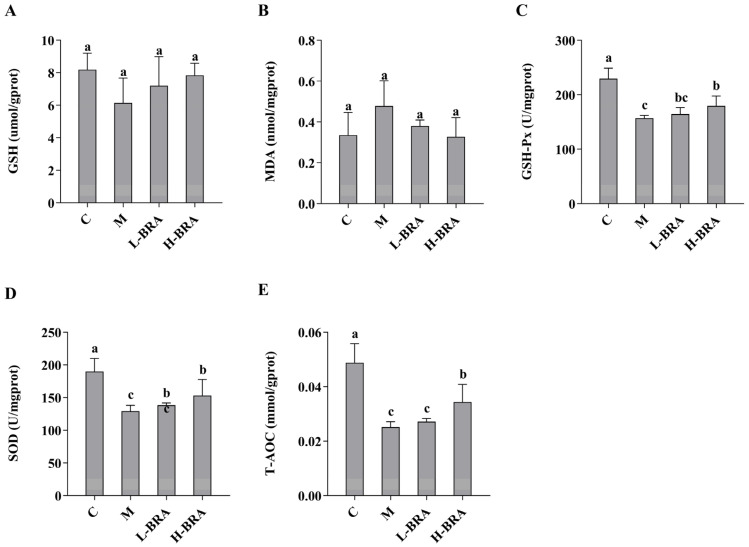
Effects of black rice anthocyanins (BRAs) on GSH, MDA, GSH-Px, SOD, and T-AOC contents in kidney tissues of aging mice induced by D-galactose (D-gal). (**A**) GSH; (**B**) MDA; (**C**) GSH-Px; (**D**) SOD;s (**E**) T-AOC. C: control group; M: D-gal model group; L-BRA: low-dose BRA group (50 mg/kg); and H-BRA: high-dose BRA group (100 mg/kg). The data are presented as the mean ± SD (*n* = 10). Different letters indicated significant differences (*p* < 0.05).

**Figure 3 nutrients-17-00502-f003:**
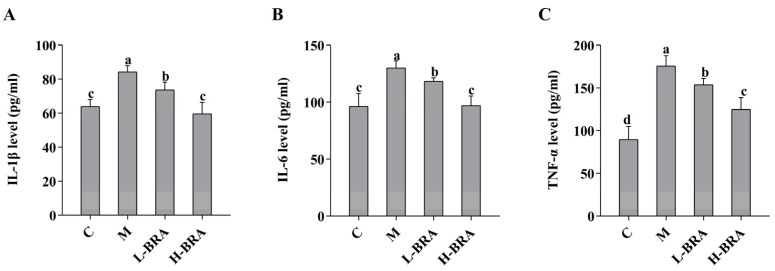
Effect of black rice anthocyanins (BRAs) on serum levels of IL-1β, IL-6, and TNF-α in aging mice induced by D-galactose (D-gal). (**A**) IL-1β; (**B**) IL-6; (**C**) TNF-α. C: control group; M: D-gal model group; L-BRA: low-dose BRA group (50 mg/kg); and H-BRA: high-dose BRA group (100 mg/kg). The data are presented as the mean ± SD (*n* = 10). Different letters indicated significant differences (*p* < 0.05).

**Figure 4 nutrients-17-00502-f004:**
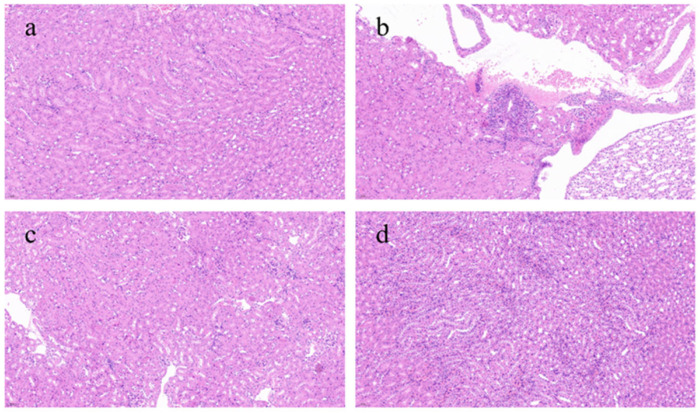
Effects of black rice anthocyanins (BRAs) on renal tissue microstructure of aging mice induced by D-galactose (D-gal). (**a**) control group; (**b**) D-gal model group; (**c**) low-dose anthocyanin group (50 mg/kg); and (**d**) high-dose anthocyanin group (100 mg/kg).

**Figure 5 nutrients-17-00502-f005:**
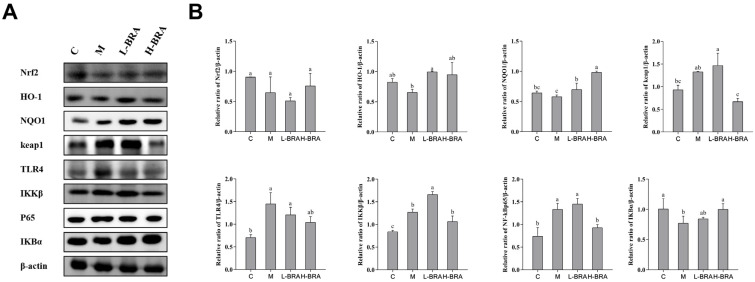
The impact of black rice anthocyanins (BRAs) on Nrf2 and NF-κB signaling pathways within the kidney tissues of aging mice subjected to D-galactose (D-gal) induction. (**A**) Western-blot results of Nrf2, HO-1, NQO1, keap1, TLR4, IKKβ, P65 and IκBα, with β-actin as a loading control; (**B**) Relative band intensities of Nrf2, HO-1, NQO1, keap1, TLR4, IKKβ, P65 and IκBα. C: control group; M: D-gal model group; L-BRA: low-dose BRA group (50 mg/kg); and H-BRA: high-dose BRA group (100 mg/kg). The data are presented as the mean ± SD (*n* = 3). Statistical significance was denoted by different letters, indicating differences at *p* < 0.05.

**Figure 6 nutrients-17-00502-f006:**
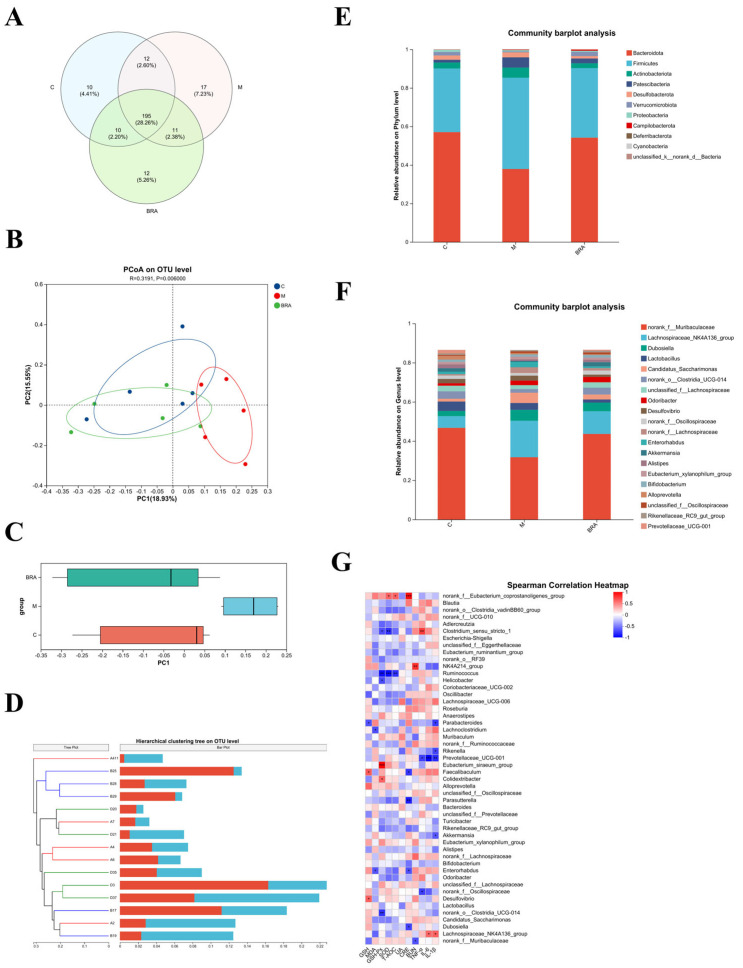
Results of mouse gut microflora analysis. (**A**) Venn diagram of mouse gut microorganisms. (**B**,**C**) PCoA of the gut microorganisms at the Operational Taxonomic Unit (OUT) level (utilizing the Bray_Curtis distance algorithm). (**D**) Hierarchical clustering tree of mouse gut microorganisms at the OUT level (based on Bray_Curtis distance algorithm). (**E**) Relative abundance of mouse gut microorganisms at the phylum level. (**F**) Relative abundance of mouse gut microorganisms at the genus level. (**G**) Spearman’s correlation analysis of gut microbiota genera in mice with physiological indices. C: control group; M: D-gal modeling group; and BRA: high-dose BRA group (100 mg/kg). Distinct letters signify statistically significant differences (*: *p* ≤ 0.05; **: *p* ≤ 0.01; ***: *p* ≤ 0.001).

**Table 1 nutrients-17-00502-t001:** Alpha diversity index of mouse gut microorganisms at the OUT level.

Sample/Estimators	Ace	Chao	Shannon	Simpson
C	523.45 ± 26.54 ^ab^	521.73 ± 27.20 ^ab^	4.35 ± 0.13 ^ab^	0.02 ± 0.00 ^a^
M	470.81 ± 51.58 ^b^	472.46 ± 47.42 ^b^	4.00 ± 0.22 ^b^	0.05 ± 0.02 ^a^
H-BRA	582.53 ± 39.56 ^a^	570.45 ± 36.00 ^a^	4.41 ± 0.10 ^a^	0.03 ± 0.01 ^a^

C: control group; M: D-gal model group; and H-BRA: high-dose BRA group (100 mg/kg). The data are presented as the mean ± SD (*n* = 5). Distinct letters signify statistically significant differences (*p* < 0.05).

## Data Availability

The original contributions presented in this study are included in the article. Further inquiries can be directed to the corresponding author.

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
