# Peer review of "Protective Effects of Black Rice Anthocyanins on D-Galactose-Induced Renal Injury in Mice: The Role of Nrf2 and NF-κB Signaling and Gut Microbiota Modulation"

_nutrients, 2025, doi:10.3390/nu17030502_

Round 1

Reviewer 1 Report

Comments and Suggestions for Authors

The current manuscript presents the findings of a study in mice designed to evaluate black rice anthocyanins in attenuation of kidney injury in a model of aging/kidney disease. The study is interesting, but the manuscript needs additional details, clarity, and other revisions.

Lines 39-43: The statements regarding "chemical drugs" for kidney disease prevention, and also for causing kidney injury is vague and should be more specific. What chemical drugs are used to prevent kidney disease?

Authors state on lines 43-44 that development of food-derived bioactive components for prevention of kidney disease is a trend, but there is no reference.

Lines 47-48: Also vague with no detail of demonstrated effects.

Lines 50-67: Unclear why the focus is on other studies not investigating the outcomes evaluated in the current study. The kidney does not come into the paragraph until line 58. May be best to summarize the earlier statements and provide the detail only as relevant.

Please state in lines 81-93 whether assays were performed in duplicate or triplicate (or single) and whether manufacturer instructions were followed. Please also specify for other assays performed in different sections if not already specified.

Lines 96-115: Please provide a reference for the model and methods used here.

Details regarding methods of euthanization are lacking on line 139.

Line 188-189: Unclear what it means that "all experimental results were repeated more than three times" - please clarify. If everything was performed in triplicate it should be specified in each relevant section.

Please specify how the anthocyanin doses were determined - based on calculations for human relevance and/or previous studies? 

Was the anthocyanin supplement independently tested to ensure contents? Please specify if so or list as a limitation of the study.

Figure 6 is difficult to read and small.

The discussion of previous research as it relates to the current finding is vague within the discussion section. This would benefit from expanding upon. Additionally, the relationship of many outcomes (inflammation, antioxidants, gut microbiome, etc.) with renal injury with aging is lacking in the discussion. The connections should be better explained. The way it is currently written they are mainly described as separate processes that occur with aging.

Please discuss how the dose provided relates to human ingestion/supplementation. Are there studies in humans that can be discussed as well? Is the dose a feasible and safe human dose? 

A limitations section needs to be added to the discussion.

Comments on the Quality of English Language

Can be improved significantly.

Author Response

Comments 1: Line 39-43 - The statements regarding "chemical drugs" for kidney disease prevention, and also for causing kidney injury is vague and should be more specific. What chemical drugs are used to prevent kidney disease?

Response 1: Thanks very much for your comments. According to your suggestion, we've made the following changes. Nowadays, chemical drugs are mainly used for the prevention and management of kidney-related diseases, but they usually have certain side effects[3]. For example, angiotensin-converting enzyme inhibitors (ACEIs), angiotensin II receptor blockers (ARBs), and statins[3,4], etc. Please see lines 41-42.

Comments 2: Authors state in lines 43-44 that the development of food-derived bioactive components for the prevention of kidney disease is a trend, but there is no reference.

Response 2: Thanks very much for your comments. According to your suggestion, we've revised this section and deleted the sentence. Please see lines 43-55.

Comments 3: Lines 47-48: Also vague with no detail of demonstrated effects.

Response 3: Thanks very much for your comments. According to your suggestion, we've made the following changes. Studies have also shown that natural bioactive substances such as resveratrol, curcumin, puerarin and anthocyanins possess therapeutic potential in alleviating acute kidney injury and diabetic kidney disease injury[7-10]. Please see lines 48-50.

Comments 4: Lines 50-67: Unclear why the focus is on other studies not investigating the outcomes evaluated in the current study. The kidney does not come into the paragraph until line 58. May be best to summarize the earlier statements and provide the details only as relevant.

Response 4: Thanks very much for your comments. According to your suggestions, We've revised our manuscript carefully, see lines 61-68.

Comments 5: Please state in lines 81-93 whether assays were performed in duplicate or triplicate (or single) and whether manufacturer instructions were followed. Please also specify for other assays performed in different sections if not already specified.

Response 5: Thanks very much for your comments. According to your suggestion, we've made the We have made the corresponding modifications. See lines 97-100.

Comments 6: Lines 96-115: Please provide a reference for the model and methods used here.

Response 6: Thanks very much for your comments. According to your suggestion, the relevant references have been added to our manuscript, see lines 113-118.

Comments 7: Details regarding methods of euthanization are lacking on line 139.

Response 7: Thanks very much for your comments. According to your suggestion, we've made the following changes. Following the euthanasia of the mice via cervical dislocation, the kidney tissues were quickly removed and immersed in a 4% paraformaldehyde solution for 24 hours to preserve the structural integrity of the tissues. See line 148.

Comments 8: Line 188-189: Unclear what it means that "all experimental results were repeated more than three times" - please clarify. If everything was performed in triplicate it should be specified in each relevant section.

Response 8: Thanks very much for your comments. According to your suggestion, we've deleted the sentence “All experimental results were repeated more than three times”. Additionally, the number of biological replicates for each experiment has been indicated in the figure captions.

Comments 9: Please specify how the anthocyanin doses were determined - based on calculations for human relevance and/or previous studies.

Response 9: Thanks very much for your comments. The dosage of anthocyanins is determined based on previous studies, and the relevant reference has been cited in the main text. See line 118.

Comments 10: Was the anthocyanin supplement independently tested to ensure contents? Please specify if so or list it as a limitation of the study.

Response 10: Thanks very much for your comments. The anthocyanin supplements we used were purchased from Nanjing Jingzhu Biotechnology Co., Ltd., and the certificate of analysis was provided by the company. The purity of cyanidin-3-glucoside (C3G) was tested to be 94.59%. The details are shown in the MS word.

Comments 11: Figure 6 is difficult to read and small.

Response 11: Thanks very much for your comments. We've readjusted the clarity of the image.

Comments 12: The discussion of previous research as it relates to the current finding is vague within the discussion section. This would benefit from expanding upon. Additionally, the relationship of many outcomes (inflammation, antioxidants, gut microbiome, etc.) with renal injury with aging is lacking in the discussion. The connections should be better explained. The way it is currently written they are mainly described as separate processes that occur with aging.

Response 12: Thanks very much for your comments. According to your suggestion, we've revised the conclusion carefully. Please see the discussion section in the revised manuscript.

Comments 13: Please discuss how the dose provided relates to human ingestion/supplementation. Are there studies in humans that can be discussed as well? Is the dose a feasible and safe human dose?

Response 13: Thanks very much for your comments. In this study, the doses of black rice anthocyanins used for feeding mice were 50 mg/kg and 100 mg/kg, which are consistent with the doses reported in most literature. Based on the body surface area method, the dosing in mice is approximately 9.1 times higher than that in humans when calculated per unit body weight. Therefore, the doses used in this study are equivalent to 5.5 mg/kg and 11.0 mg/kg in humans. The Joint Expert Committee on Food Additives (JECFA) of the World Health Organization (WHO) and the Food and Agriculture Organization (FAO) has established the Acceptable Daily Intake (ADI) of anthocyanins for humans at 0–2.5 mg/kg body weight. Although no clear adverse effects of anthocyanin intake have been identified in humans or animals to date, it is recommended that the daily intake should not exceed 800 mg. For an adult weighing 70 kg, this is equivalent to 11.43 mg/kg. The doses used in this study did not exceed the tolerable levels for humans. Thus, it is suggested to be a safe human dose.

However, by reviewing the literature, I found that there are few studies on the relationship between the dosage of black rice anthocyanins and human intake or supplementation, and there is also a lack of relevant human studies.

In summary, although the doses used in our study are higher than the daily intake levels in humans, they have demonstrated good protective effects in animal experiments. According to existing data from human studies, the intake and supplementation of anthocyanins are safe and feasible, and significant health benefits have been shown even at lower doses. Future research can further explore the long-term effects of anthocyanins at different doses to optimize their application in human health.

Comments 14: A limitations section needs to be added to the discussion.

Response 14: Thanks very much for your comments. We have revised our manuscript according to your suggestion, see lines 467-476.

Reviewer 2 Report

Comments and Suggestions for Authors

The submission reports a study on the protective effects of black rice anthocyanins.

The experimental design is well structured and rigorously conducted.

The results are commented and supported by statistical evaluation.

Images are also attached as supplementary material, but they are not of high quality.

I suggest a minor revision, mainly on the following comments:

- authors should justify their choice to use only male mice in section 2.2

- authors should better explain how " the mice were humanely euthanized" to ensure that no influence was induced on the following experiments

- figure 6 is complete but, in my opinion, not so easy to read because the characters are very small; is it possible to improve the figure? Maybe by enlarging the size or dividing it...

Author Response

Comments 1: authors should justify their choice to use only male mice in section 2.2

Response 1: Thanks very much for your comments. Only male mice are used in animal experiments for the following reasons: the physiological characteristics of male mice are relatively homogeneous, which can reduce the fluctuation of experimental results caused by sex differences, and make the data more stable and reproducible; The hormone level in the body fluctuates little, which can avoid the interference of hormonal changes brought about by the estrous cycle of female mice to the experimental results. In the construction of some disease models, the pathogenesis and process of male mice are relatively typical, which is conducive to the study of the core mechanism of disease. At the same time, male mice have traditionally been used in some research areas, which is convenient for comparison and integration with previous studies, and can reduce the difficulty of complex data analysis due to sex differences.

Comments 2: authors should better explain how " the mice were humanely euthanized" to ensure that no influence was induced on the following experiments

Response 2: Thanks very much for your comments. According to your suggestion, we've made the following changes. Following the euthanasia of the mice via cervical dislocation, the kidney tissues were quickly removed and immersed in a 4% paraformaldehyde solution for 24 hours to preserve the structural integrity of the tissues. See line 148.

Comments 3: figure 6 is complete but, in my opinion, not so easy to read because the characters are very small; is it possible to improve the figure? Maybe by enlarging the size or dividing it...

Response 3: Thanks very much for your comments. We've readjusted the clarity of the image.

Reviewer 3 Report

Comments and Suggestions for Authors

The manuscript, nutrients-3408664, is dedicated to the evaluation of the renal protective effects of black rice anthocyanins against renal injury using a murine model of aging, established by subcutaneously injecting D-gal to the dorsal cervical region of mice.

Anthocyanins belong to compounds that have been studied for medicinal purposes for a long time. The application of modern experimental techniques is a step forward in understanding the therapeutic effects of anthocyanins.

Despite the fact that the obtained results are the object of deeper discussion, in my opinion, the presented study delivers original experimental data that will be interesting to readers. My comments mostly related to technical corrections of the manuscript (below).

Comments:

  1. Please increase the line thickness and text size in the figures.
  2. I suggest extending the introduction parts with wider analysis of the utilization of anthocyanins for therapeutic purposes. For example,

DOI: 10.3390/plants13223136

DOI: 10.1002/jsfa.7181

DOI: 10.1016/j.intimp.2024.113385

DOI: 10.3390/toxins16030151

Author Response

Comments 1: Please increase the line thickness and text size in the figures.

Response 1: Thanks very much for your comments. Based on your suggestion, we have made changes to the main text.

Comments 2: I suggest extending the introduction parts with a wider analysis of the utilization of anthocyanins for therapeutic purposes. For example,

DOI: 10.3390/plants13223136

DOI: 10.1002/jsfa.7181

DOI: 10.1016/j.intimp.2024.113385

DOI: 10.3390/toxins16030151

Response 2: Thank you very much for your valuable suggestions. According to your comments, we have expanded the analysis on the therapeutic applications of anthocyanins in the Introduction section and have added relevant references. See lines 50-55.